# The Influence of Previous Lifestyle on Occupational Physical Fitness in the Context of Military Service

**DOI:** 10.3390/ijerph20031860

**Published:** 2023-01-19

**Authors:** Leila Oja, Jaanika Piksööt

**Affiliations:** National Institute for Health Development, 11619 Tallinn, Estonia

**Keywords:** APFT, health behavior, mandatory military service, physical activity, physical fitness

## Abstract

The Estonian Defense Forces are the basis of military service, mandatory for all male citizens of the Republic of Estonia who are at least 17 years old. The physical load in military service, especially in the first stage, is significantly greater than for men in everyday life. Therefore, it is important to know if health promotion in civilian life adequately prepares young people for military service and to what extent pre-military health behaviors affect physical performance during service. The purpose of this work was to examine conscripts’ physical fitness at different stages of military service and its relationships with previous lifestyle. Soldiers’ physical fitness was estimated three times during military service using three tests: sit-ups, push-ups and 2-mile run. Lifestyle and socio-economic background data was collected by a web-based questionnaire (n = 235). Linear regression analysis was performed using Army Physical Fitness Test (APFT) scores as dependent variables and questionnaire data as independent variables. The socio-economic background variables had no effect on physical fitness scores throughout the training period (*p* > 0.05). Young men that were physically more active daily, did sports, had healthier diet and did not smoke before entering military service showed better physical fitness test results throughout the period of service (*p* < 0.05). The effect of participation in sports was evident, as the conscripts with previous sports experiences demonstrated higher fitness tests scores (*p* < 0.01). These findings show that health promotion initiatives or programs for promoting physical activity and healthy diet, and preventing obesity and tobacco use, can also have a positive effect on the physical performance of young men during military service.

## 1. Introduction

The military capacity in terms of human resource of member states of the European Union is mostly based on professional soldiers. Estonia is one of the eight countries in the EU, along with Austria, Cyprus, Denmark, Finland, Greece, Lithuania, and Sweden, where mandatory military service has been maintained, or conscription very recently re-introduced conscription [1].

In the Estonian Defense Forces, military service or alternative civil service (for religious or moral reasons) is compulsory for all 17–27 year old male citizens [2]. The duration of the compulsory military service is 8 or 11 months, depending on the education and position provided by the Defense Forces to the conscript. Prior to military service, young people undergo compulsory medical examinations. Young men who have been attested to be healthy by the medical commission have to enter military service. The Soldier’s Basic Course (SBC) continues up to 12 weeks. After the SBC, the Soldier Specialty Basic Course (SSBC) follows; the length of the course depends on the complexity of the position (specialty) of the conscript. The conscripts’ physical fitness is evaluated before and after the SBC and for the third time before the end of military service using the Army Physical Fitness Test (APFT) [3]. The assessment of physical fitness is organized according to standard procedures in the Estonian Defense Forces.

Conscripts’ health and physical capabilities are essential to successfully complete the training and thus create a competent reserve army. It is important to find out the factors that contribute to young people’s physical capabilities in order to proactively assess and find solutions to develop their physical capabilities and maintain the level reached after military service and when acting as a reserve.

Monitoring the physical fitness level of the military and security forces is important from a performance point of view, as well as to assess combat ability [4]. Inadequate fitness as a problem has been confirmed by previous studies. Low physical activity and physical fitness are followed by an increased incidence of diabetes mellitus and nonfatal cardiovascular diseases [5]. Low levels of physical fitness are strongly associated with proneness toward overuse injuries in soldiers going through intense training [6]. A higher physical fitness score is associated with a healthier CHD risk factor profile and is a predictor of better pre-deployment cardiovascular health [7].

Physical activity as an important component of military recruitment has also been evaluated in the US military. Along with BMI, low levels of physical activity have been identified as a potential threat to US national security, as they affect military recruitment [8]. During the last decade, only 17–21% of Estonian adolescent boys have been active daily, with moderate to vigorous physical activity for at least 60 min a day [9]. Compared to other countries in the European region [10], the physical activity of Estonian youth is below average level. Therefore, there is a reason to believe that most young boys recruited for military service will probably experience the effects of systematic and strong physical training for the first time in their lives at the beginning of military service, during the soldier’s basic course. Research based on Finnish soldiers demonstrated that low physical performance is one important reason for dropout before the end of basic training course, due to health problems or injuries [11]. In 2016, almost 20% of Estonian conscripts were prematurely discharged from military service every year due to health problems, including injuries and diseases of the musculoskeletal system [12]. In recent years, the drop-out rate has decreased. In 2019 the drop-out rate was 7.8%, then in 2020 was only 5.6% [13]. However, there is a reason to believe that since the health of young men in the population has not significantly improved, the desire of young men to contribute to national defense and acquire relevant skills has increased due to the security situation arising from a neighboring country. Despite the vast amount of knowledge available that states that adolescents partaking in regular physical activity are more likely to adopt a healthy and balanced diet that helps to improve physical and personal well-being, adolescents’ physical activity has gradually decreased [9].

Based on the literature, research has sought to find out what preparation would help young men to be in better physical shape and to end their military service without injury. Studies have explained that recruits with lower pretraining fitness levels, low body mass and past injuries were at higher risk for injuries. Soldiers should develop and maintain high levels of physical fitness, not only for optimal performance of occupational tasks but also to reduce injury risk [14]. Lifestyle indicators have been evaluated also for preventing soldiers’ injuries. The influence of lifestyle on injuries in the armed forces has been evaluated in an overview study, in which physical activity, diet and BMI are among the large variety of investigated factors [15]. It is recommended that faster run time performance and minimal body mass standards should be considered for physical entry criteria [16]. Participating for 30 min in vigorous physical activity is one of the factors of trainability and predictability associated with military physical fitness test success [17]. A higher level of physical activity helps conscripts to adapt to the new situation and to the effort involved in military service, and to reduce dropout rates. Previous studies using the example of Estonian conscripts have explained that causes for the premature dropout of conscripts from military service have not been systematically studied. One of the factors involved could be the relatively demanding physical training that starts at the beginning of military service. Therefore, most of the conscripts are likely to receive their first experience of demanding systematic physical exercise during basic military training, where the load may exceed their previous training level [18].

Different stages of military training demand different physical efforts. This is particularly intense during the first period of training (SBC). During the following periods, the focus is on more specific skills. It is important to explain from the point of practical considerations how to improve military training and does a lifestyle and physical fitness affect the performance at different stages of military training, which have different physical loads and intensity. Conscripts’ health and physical fitness are essential for completing the military training successfully and to create an efficient reserve army. Therefore, it is important to find out the factors that contribute to young people’s physical capabilities and work out solutions to support sufficient physical fitness throughout military service.

The purpose of this study is to examine male conscripts’ physical fitness at different stages of military service and its relationship with conscripts’ socio-economic background, general health characteristics and health behavior. The main emphasis is on whether the conscripts’ healthy lifestyle or participation in sports before attending the conscription influence their performance throughout the service. Although changes in physical performance of conscripts during military service have been studied in the past [19], there is little evidence about changes in physical performance related to conscripts’ previous lifestyles.

## 2. Materials and Methods

The research data included the Army Physical Fitness Test (APFT) for measuring conscripts’ physical performance and a web-based questionnaire to assess their socio-economic background, health characteristics and health behavior.

To assess soldiers’ physical performance and their suitability for training in the Estonian Defense Forces, the United States Army Physical Fitness Test (APFT) is used [3]. APFT is a compulsory fitness test for measuring the physical performance of soldiers in the Estonian Defense Forces.

APFT is a standardized measurement of physical fitness that consists of three physical exercises: (1) the number of push-ups performed in 2 min (minimum performance to pass the test is 40 times); (2) the number of sit-ups performed in 2 min (minimum performance is 46 times); (3) the amount of time required for completing a 2-mile run (minimum running time to pass the test is 15:54). The results are presented in scores calculated according to the score table. Scoring of the APFT is based on gender, age category and the number of repetitions performed. Each of these three APFT events has a maximum score of 100 points, meaning that the maximum possible score for the APFT is 300 points. A pass score for the test is a minimum of 190 points as the sum of three exercises, or at least 60 points for each event. A higher score indicates a better physical performance level.

The current research included the APFT results of 1906 male conscripts aged 18–28 (m = 21.5, SD = 1.7) from 9 Estonian contingents. In Estonia, selectees are called up for conscription three times a year. This sample involved conscripts that started the service in the summer period (July), which was the largest recruitment of the year.

During the conscripts’ military service, the APFT was performed three times: (1) at the beginning of the military service, baseline; (2) after Soldiers’ Basic Course (SBC), 12 weeks from the beginning of service; and (3) at the end of the military service, at 8 or 11 months (Figure 1). The data was collected in 2014 in cooperation with the personnel department of the Headquarters of the Estonian Defense Forces. The study was approved by The Tallinn Medical Research Ethics Committee (No. 1674).

All conscripts underwent SBC according to the standard program established by the Command of the Estonian Defense Forces which included physically demanding activities such as combat training, sports training, marching drills and marching exercises. During the SBC, in addition to compulsory physical education, conscripts also had the opportunity to practice independently in the gym. The physical training load and intensity increased, according to the training plan, from the beginning and reached its highest level at the end of SBC after 12 weeks.

A random selection, about 12% of conscripts from various military units from all over Estonia filled in a web-based questionnaire at the beginning of their military service (Figure 1). In total, the questionnaire was filled in by 235 conscripts aged 21.3 ± 1.6 years. The questionnaire was designed to assess conscripts’ socio-demographic background (age, living place, nationality, education, economic situation, occupation), general health (diagnoses, usage of medicines), and health behavior (physical activity, inc. previous sports experiences; nutrition, smoking and alcohol consumption). The questionnaire items originate from a survey Health Behavior among Estonian Adult Population, which has been conducted in Estonia each even year starting from 1990 and enables comparisons with the general population [20].

The respondents’ BMI (m = 23.5 ± 3.6 kg/m^2^) was calculated based on objectively measured weight (m = 78.9 ± 12.1 kg) and height (m = 1.83 ± 0.07 m). Conscripts’ height and weight were measured in the medical centers of the contingents at the beginning of the service (in July) by using a standard procedure. According to the initial BMI, 77% of conscripts were underweight/normal weight (BMI ≤ 24.9), 17% overweight (BMI = 25.0–29.9) and 6% obese (BMI ≥ 30.0).

For statistical analyses, IBM SPSS Statistics software version 22.0 was used (IBM Software Group, Chicago, IL, USA). Depending on the scale of variables, the data was described with means (m), standard deviations (SD), frequencies (n) and percentages (%). The changes in conscripts’ physical performance during military service were clarified by comparing APFT results with one-factor repeated measures analysis of variance (ANOVA). Bonferroni method was used for pairwise comparisons. For analyzing relationships between APFT results and self-reported data, the questionnaire and APFT databases were merged with subjects’ ID.

To examine the effect of socio-demographic background, health characteristics and health behavior on conscripts’ physical performance, linear regression analysis was performed using APFT total scores as dependent variables. To clarify the influence of these factors at the different stages of military service, regression analysis was performed on each of the three tests time results. As a result, standardized regression coefficients (β) and t-values were presented. All analyses were considered statistically significant at the level of *p* < 0.05.

## 3. Results

According to one-factor (time) ANOVA with repeated measures, the mean scores for APFT were statistically significantly different during the three measuring times (F(2, 882) = 529.6, *p* < 0.001 for number of push-ups; F(2, 882) = 262.0, *p* < 0.001 for number of sit-ups; F(2, 882) = 261.9, *p* < 0.001 for 2-mile run; F(2, 882) = 575.2, *p* < 0.001 for total score; Table 1)).

The pairwise comparisons of APFT scores indicated that the conscripts’ physical performance improved significantly during the SBC (Bonferroni method, *p* < 0.001). Only 60% of conscripts were able to pass the APFT at the beginning of military service. After the end of SBC, the proportion passing was raised to 95% (Table 1). The highest improvement occurred in the conscripts’ push-up performance: from 72% passes at the beginning of the service to 97% after SBC (*p* < 0.001). Passing for sit-ups increased from 62% to 88% and of 2-mile run from 59% to 90% (*p* < 0.001).

Comparison of mean scores of the second and the third test, on the contrary, indicated that the physical performance of conscripts had decreased by the end of the service, especially in sit-ups and 2-mile run (*p* < 0.001). Thus, the improvement of conscripts’ physical performance did not continue throughout all military service, but instead started to decrease at the later stages.

Linear regression analysis on the APFT scores was performed, using socio-economic, general health and health behavior factors as the independent variables; the descriptive data for the variables is presented in Table 2.

Results of regression analysis (Table 3) showed that none of the socio-economic factors were significantly related with the APFT scores of conscripts. Statistically significant relations, however, were found with health characteristics and health behavior.

The conscripts’ performance at the beginning of the military service (the first APFT score) was negatively related with BMI and positively related with self-assessed health and self-assessed physical performance (Table 3). From the health behavior factors, the first APFT performance was increased by higher physical activity, previous sports experiences, healthy eating and decreased by smoking. The frequency of alcohol consumption had no effect on APFT performance.

The APFT performance after the SBC (the second APFT score) was significantly related to conscripts’ BMI (measured at the beginning of service), self-assessed health and self-assessed physical performance, physical activity and healthy nutrition. The effect of smoking disappeared, so the conscripts who had been smoking before military service, had improved their performance. The effect of previous sports experiences also had no effect on the APFT performance after SBC.

At the end of the military service, the conscripts’ APFT performance (the third APFT score) was again significantly related to self-assessed health, self-assessed physical performance, non-smoking and physical activity. The effect of sports experiences also re-appeared, as the conscripts with previous sports experiences again demonstrated higher APFT scores.

## 4. Discussion

Similar to other countries [8], the behavior of Estonian schoolchildren has mostly become less healthy [9]. When the health behavior indicators of young people in the population are low and health promotion initiatives are not sufficient, it is very problematic to find young people with suitable health for military service. Soldiers are expected to maintain a high level of physical performance to be ready to mobilize and perform their duties through the whole military training period and, later, in the reserve army. This research demonstrated that, when it comes to soldiers’ physical performance, their pre-military lifestyle is crucial—daily physical activity, sports experiences, healthy diet, non-smoking and lower BMI contribute to better performance in physical fitness tests, whereas soldiers’ socio-economic background is insignificant.

The goal of military basic training is to create a foundation for physical fitness and military skills in soldiers. Thereafter, more advanced military training can safely take place [21]. Previous studies have found that low fitness levels before service were associated with training-related injuries and attrition [22,23] and these are critical issues for modern armies with increased demands for well-prepared soldiers and fewer injury losses [24]. The present study demonstrated associations between physical activity and passing of the fitness test, confirming the general fitness benefits of activity for different stages of military training. For instance, those conscripts who were physically active at least two times a week before conscription performed APFT tests at a higher level during all stages of military training.

Military fitness studies have confirmed that, on average, the physical fitness of conscripts improved during their compulsory military service. Conscripts with a lower baseline fitness level or higher BMI showed the largest improvements, which may be a significant finding from both a military readiness and national health perspective [19]. Our results showed, that before entering the military service, the conscripts’ physical fitness level was quite low. Minimum requirements were achievable for only slightly more than half of the conscripts (60% passed the test positively). After 12 weeks of military training, up to 95% of conscripts were able to pass the APFT. The analysis showed that socio-economic background variables were not related to APFT scores, but general health and health behavior indicators had a significant impact on conscripts’ physical fitness.

Common health and lifestyle risk factors for premature discharge or injury are mostly lack of practice of competitive exercises [25,26], poor self-rated physical fitness [25,27], low self-assessed health [11,28], low pre-service physical activity [27,29] and low level of physical fitness [11,14,16,23]. A very thorough overview of lifestyle factors that specifically influence musculoskeletal injuries has been made by Sammito and co-authors [15]. In this analysis, it was found that, after basic training, conscripts’ self-assessed health and self-assessed physical performance, physical activity and healthy nutrition were significantly related to APFT performance. Previous sports experiences had no effect after basic training, but at the end of the military service the effect reappeared, as the conscripts with previous sports experiences performed higher in APFT. Analyses have shown that conscripts in a good physical condition can also improve this during their military service. However, the decline in physical performance of highly-fit conscripts highlights the importance of individualization of physical training and military training load during military service [19]. After the end of SBC, the conscripts were no longer provided with organized activities and guidance to maintain physical fitness; instead, they had to practice more independently, where previous sports experience proved beneficial.

We found that, after the SBC period, BMI was one of the factors influencing soldiers’ APFT performance, which is in good accordance with recent findings [30]. The objective of army body composition standards is to motivate physical training and good nutrition habits to ensure a high state of readiness [31]. In the case of compulsory military service, conscripts are ordinary young people who are a part of the normal population. The Estonian Defense Forces Medical Service uses BMI to determine the obesity of conscripts. BMI has been the most frequently reported anthropometrical measure to be independently associated with injury risk [16] and increased body weight was associated with declined performance on endurance run [32]. A recent study [30] provided a valuable insight into the topic, showing that the performance of overweight soldiers can be improved by adding regular strength training to their daily physical training.

This analysis showed that military training was more successful for those conscripts whose health indicators were better, who were physically active daily before entering military service and who had a healthy lifestyle. We found that conscripts’ self-assessed health and self-assessed physical performance were positively related with APFT scores throughout the training period. The use of questionnaires as a preventive approach is also suggested by previous studies. For example, close correlation between the actual and self-reported scores suggest that self-reported values are adequate for most epidemiological military studies involving large sample sizes [33]. It has also been found that recruiters could possibly use the questionnaire to identify those who should participate in additional preliminary physical training, while policy makers may consider using it to identify those who should undergo either closer preinduction examination or postinduction remedial training during later training [22].

Surveys based on self-assessed short questionnaires should include an evaluation of activity questions, validation of a fitness test and measures of outcome, such as training success or risk of injury. These reports provide information that may be useful to military recruiters and policy makers, as this brief questionnaire identifies those at increased risk of failing to pass a simple fitness test, and failing the test is a significant predictor of musculoskeletal injury [22]. Our research also verified that questioning the conscripts before military service would help anticipate physical training success or risk of injury for the whole training period.

Limitations of the study include the usage of self-reported data which may cause some bias due to social desirability, inaccurate recall, misinterpretation of terms, etc. The response rate was also quite low, which sets limits to generalizability of the findings. Additionally, the findings of the current study cannot be generalized, as the army has many different branches with various tasks that need specific levels of physical abilities and combat readiness.

## 5. Conclusions

The results of this study show a predictive association between health behavior indicators and APFT scores. The soldiers who before entering military service were physically more active daily, did sports, ate healthier food and did not smoke showed better physical test results throughout their military service. Preventive measures and programs are clearly needed and, optimally, should be tested in controlled intervention studies [11].

In practice, these findings suggest that using health behavior questionnaires as tools and results as predictors of conscripts’ physical performance, we can divide conscripts into different fitness groups at the very beginning of their military service to provide instructions adapted to individual physical abilities. The results show that, in civilian life, before military service, health promotion in terms of physical activity, smoking and healthy diet influence the physical performance of young men during military service.

## Figures and Tables

**Figure 1 ijerph-20-01860-f001:**
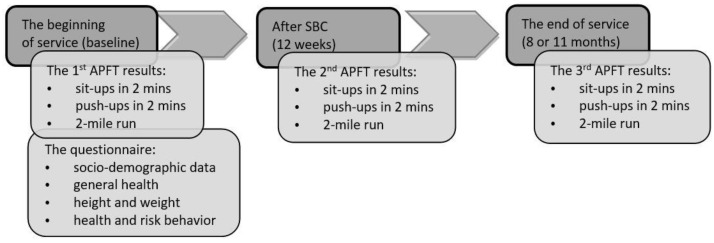
The data collection during military service (SBC—Soldier’s Basic Course; APFT—Army Physical Fitness Test).

**Table 1 ijerph-20-01860-t001:** The average results of conscripts’ (n = 883) APFT events and the overall score; the proportion of conscripts that passed the test in each measurement time.

APFT Event	APFT I (Baseline)			APFT II (after 12 Weeks)			APFT III (8 or 11 Months)			F(2, 882) ^a^
	m ± SD	Min–Max	Passed (%)	m ± SD	Min–Max	Passed (%)	m ± SD	Min–Max	Passed (%)	
Number of push-ups	48.8 ± 17.0	2–129	72%	69.4 ± 15.6	3–143	97%	68.2 ± 14.3	20–133	98%	529.6 *
Number of sit-ups	50.8 ± 17.5	4–118	62%	62.7 ± 14.7	17–125	88%	58.0 ± 14.0	13–107	81%	262.0 *
2-mile run (minutes)	15:50 ± 02:36	10:52–32:00	59%	14:00 ± 01:31	10:07–21:53	90%	14:46 ± 01:52	10:41–38:59	80%	261.9 *
Total score	198.6 ± 57.3	0–300	60%	255.2 ± 36.9	79–300	95%	242.5 ± 35.9	108–300	93%	575.2 *

^a^ Repeated measures ANOVA, * Sig. < 0.001.

**Table 2 ijerph-20-01860-t002:** Questionnaire data of the conscripts’ (n = 235) socio-economic background, subjective general health and health behavior characteristics.

Variables	Item	Answer Categories	n (%)
Socio-economic background	Living place	City/Town	155 (66.0%)
		Countryside	80 (34.0%)
	Nationality	Estonian	211 (89.8%)
		Non-Estonian	24 (10.2%)
	Family’s economic situation	Very bad	4 (1.7%)
		Bad	38 (16.2%)
		Sufficient	93 (39.6%)
		Good	100 (42.6%)
	Education level	Primary or basic education	11 (4.7%)
		Secondary education	99 (42.2%)
		Vocational secondary education	103 (43.8%)
		Higher education	22 (9.4%)
General health	Self-assessed health	Poor	6 (2.6%)
		Sufficient	34 (14.5%)
		Good	133 (56.6%)
		Very good	62 (26.4%)
	Self- assessed physical performance	Very poor	4 (1.7%)
		Poor	25 (10.6%)
		Sufficient	98 (41.7%)
		Good	91 (38.7%)
		Very good	17 (7.2%)
Health behavior	Physical activity during past 12 months (at least 0.5 h per time)	Less or never	77 (32.8%)
		Once a week	40 (17.0%)
		2–3 times a week	65 (27.7%)
		4–6 times a week	37 (15.7%)
		Every day	16 (6.8%)
	Has previous sports experiences	Yes	70 (29.8%)
		No	165 (70.2%)
	Eating habits during past 12 months	Have not thought about healthy eating	20 (8.5%)
		Do not know how to eat healthy	16 (6.8%)
		Knew how to eat healthy, but ate everything	133 (56.6%)
		Ate mostly healthy	58 (24.7%)
		Ate healthy	8 (3.4%)
	Smoking	Daily	87 (37.0%)
		Occasionally	34 (14.5%)
		Have quit	27 (11.5%)
		Have never smoked	87 (37.0%)
	Alcohol consumption during past 12 months	5–7 times a week	8 (3.4%)
		3–4 times a week	17 (7.2%)
		1–2 times a week	100 (42.6%)
		1–3 times a month	85 (36.1%)
		Less or never	25 (10.6%)

**Table 3 ijerph-20-01860-t003:** The influence of socio-economic background, health characteristics and health behavior on conscripts’ (n = 235) APFT total score according to linear regression analysis.

Model Variables	APFT Score I			APFT Score II			APFT Score III		
	β	T	Sig.	β	T	Sig.	β	T	Sig.
**Socio-economic background**									
Age	−0.04	−0.37	n.s.	−0.06	−0.52	n.s.	−0.09	−0.64	n.s.
Living place (City vs countryside)	0.06	0.51	n.s.	−0.04	−0.30	n.s.	−0.09	−0.99	n.s.
Nationality (Estonian vs other)	−0.04	−0.31	n.s.	−0.09	−0.76	n.s.	−0.01	−0.15	n.s.
Economic situation	0.04	0.32	n.s.	−0.06	−0.51	n.s.	−0.01	−0.13	n.s.
Education level	0.01	0.12	n.s.	−0.10	−0.81	n.s.	0.00	−0.03	n.s.
**General health**									
BMI ^a^	−0.25	−2.23	<0.05	−0.28	−2.34	<0.05	−0.27	−2.00	n.s.
Self-assessed health	0.49	4.85	<0.001	0.51	4.81	<0.001	0.30	3.35	<0.001
Self-assessed physical performance	0.62	6.79	<0.001	0.57	5.61	<0.001	0.54	6.93	<0.001
**Health behavior**									
Smoking (Yes vs. No)	−0.25	−2.23	<0.05	−0.18	−1.44	n.s.	−0.19	−2.06	<0.05
Alcohol consumption	0.17	1.46	n.s.	0.13	1.08	n.s.	−0.02	−0.17	n.s.
Physical activity	0.48	4.77	<0.001	0.40	3.51	<0.001	0.36	4.14	<0.001
Sports experiences (Yes vs No)	0.39	3.63	<0.001	0.22	1.83	n.s.	0.28	3.16	<0.01
Healthy eating	0.24	2.10	<0.05	0.26	2.19	<0.05	0.18	1.96	n.s.

^a^ Objectively measured BMI at the beginning of military service.

## Data Availability

Data sharing not applicable due to ethical restrictions.

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
