# Peer review of "The Influence of Previous Lifestyle on Occupational Physical Fitness in the Context of Military Service"

_ijerph, 2023, doi:10.3390/ijerph20031860_

Round 1

Reviewer 1 Report

The manuscript submitted for publication in IJERPH by Leila Oja and Jaanik Piksööt entitled: "The influence of previous lifestyle on occupational physical fitness in the context of military service" assessed the physical fitness of conscripts at various stages of military service and its relationship to the socio-economic background of conscripts , general health characteristics and health behaviors. The manuscript is well written and is well organized. It is a good read and deals with an interesting topic that is usually under-researched.

Some comments and suggestions:

·       - It is worth adding in the introduction information why physical activity among young people in Estonia is falling so much.

·      -  The methodology should better describe how conscripts were included in the study, were they all conscripts who applied from January to December 2014? This should be specified.

·       = In the introduction, the authors write that military service is compulsory for all citizens aged 17-27 And the research group is in the 18-28 age range, where does this difference come from?

·       - There is no information in the research methodology whether the online questionnaire was developed by the authors or came from other sources. It's worth supplementing.

·       - Was the deterioration of fitness test scores between APFT 2 and APFT 3 unrelated to changes in the lifestyle of the subjects during this period? Nothing is known in the manuscript about this. The authors devoted little attention to this observation also in the discussion of why the physical fitness decreased in the subjects between APFT 2 and APFT 3, it would be worth supplementing it.

·      -  Were the results of fitness tests correlated with the results of medical examinations that all conscripts underwent, as the authors wrote in the introduction?

·     -   On what basis did the subjects declare that they eat healthy or not? Did they have to meet any criteria? Applies to the results presented in Table 2.

·       - In the discussion on line 268 the authors write that the Estonian Defense Forces Medical Service uses BMI to determine the body composition of conscripts.

·       - In the discussion in line 268, the authors write that the Medical Service of the Estonian Armed Forces uses BMI to determine the body composition of conscripts. Did the authors make a mistake here? The bioimpedance method is used to assess body composition, certainly not BMI, which does not determine body composition, hence its limitations.

·       - In the results of the study, the authors did not present the percentage of overweight and obese conscripts, it is worth supplementing it. We only know the average value of the initial BMI of all the subjects. There is also no data on whether the BMI value of the subjects changed significantly during the military training. This is important because the authors correlate BMI with the results of fitness tests.

Reviewer 2 Report

Dear Authors

You have written an interesting paper examining physical fitness at different stages of military service and its relationships with the lifestyles before entering the service.

There are some parts of the paper that need to be addressed for greater clarity.

Abstract - please add p values of significant findings. The total value of tested participants is misleading as you had only 235 questionnaires filled. This needs to be clear. Please correct.

Overall the paper recently published on Risk factors for musculoskeletal injuries in the military (https://mmrjournal.biomedcentral.com/articles/10.1186/s40779-021-00357-w) could tremendously benefit your introduction and especially in the discussion part. Therefore, I recommend authors read it and amend the intro and discussion part.

Otherwise, the main rationale of the study is well presented.

Methods:

Please add the questionnaire as supplemental material. Was this questionnaire validated? Was it used before? Please report this and add possible references.

Line 154 - Please report how were the height and weight measured

Overall I see an issue as you are mixing 2 datasets. You should report only the APFT data from the participants that filled out the questionnaire at the beginning of service. In tables please be specific and add the sample size (n=xx) in the title caption or in the table. 

Was table 1 done on the whole sample of 1906 participants?

This is unclear in the paper.

Discussion

The first paragraph should present the research findings in non-statistical terms. Please amend

The paragraph in Lines 264 - 271 / a recently published paper (https://militaryhealth.bmj.com/content/168/2/141) could strengthen your BMI discussion as it confirms your findings and also adds additional solutions.

Limitations paragraph - I would recommend adding something in this form:  also the findings can't be generalised as the army has many different branches that have various different tasks that need different levels of physical abilities and combat readiness.

Overall the paper looks promising. However, the authors need to address the highlighted questions for greater clarity. Therefore, I recommend a major revision.

Kind regards

Round 2

Reviewer 2 Report

Dear Authors,

Thank you for addressing all raised questions adequately. The manuscript quality has improved. 

I would only suggest adding a conclusion sentence at the end of the abstract so you don't finish with the results sentence.

Therefore, I recommend acceptance after a minor revision.

Kind regards

Author Response

Dear Reviewer,

Thank you for the suggestion. We have added a conclusion sentence at the end of the abstract.

Kind regards,

Authors